# Effect of Organosilicon Self-Assembled Polymeric Nanolayers Formed during Surface Modification by Compositions Based on Organosilanes on the Atmospheric Corrosion of Metals

**DOI:** 10.3390/polym14204428

**Published:** 2022-10-20

**Authors:** Maxim Petrunin, Alevtina Rybkina, Tatyana Yurasova, Liudmila Maksaeva

**Affiliations:** Frumkin Institute of Physical Chemistry and Electrochemistry, Russian Academy of Sciences, 119071 Moscow, Russia

**Keywords:** organosilanes, self-assembled siloxane polymeric layers, metal corrosion, natural corrosion tests, corrosion inhibition

## Abstract

Reducing the risks caused by losses due to the atmospheric corrosion of metal structures has been relevant for many years and is an important scientific and technical task. Previously, for this purpose, the preliminary modification of the surface of structural metals with solutions of compositions, based on both individual organosilanes and their mixtures with amine-containing corrosion inhibitors, was proposed. Such treatment leads to the formation of self-assembled siloxane polymeric/oligomeric nanoscale layers on the metal surface, which are capable of changing the physicochemical properties of the metal surface (namely, by reducing the tendency of the metal to corrosive destruction). In this work, annual atmospheric corrosion tests of samples of steel, copper, zinc, and aluminum without protection, and samples modified with compositions based on organosilanes in an urban atmosphere, were carried out. It was established (by the gravimetric method) that the corrosion rate of unmodified (without protection) metals is as follows: steel—0.0022 mm/year; aluminum—0.0015 mm/year; copper—0.00018 mm/year; and zinc—0.00023 mm/year. Using gravimetry and optical microscopy, it was shown that the preliminary modification of metal surfaces with compositions based on organosilanes led to the inhibition of both uniform and local corrosion of metals. The corrosion rates of samples that were modified with one-component compositions decreased by almost two times. The maximum inhibitory effect for the studied systems was demonstrated by mixed binary modifying compositions: mixtures of vinyl- and aminosilane, vinylsilane, and benzotriazole. The corrosion rate decreased for all the studied metals. The minimum effect was observed on zinc (2.5 times) and the maximum inhibition of the corrosion rate was obtained on copper (5.1 times). The mechanism of corrosion inhibition by layers formed as a result of surface modification with two-component mixtures was considered.

## 1. Introduction

Currently, metals remain the main structural material. However, with all their advantages, they have a very important drawback: they are destroyed under the influence of the environment, i.e., they are subject to corrosion. Therefore, the fight against the corrosion of metals is an important and urgent scientific and technical problem. Atmospheric corrosion is one of the most common types of metal destruction, since a significant part of metal structures and structures is operated in atmospheric conditions. Atmospheric corrosion is understood as the destruction of metals in the air with physical and chemical parameters inherent in the real atmosphere [1]. Almost all metal structures operated in natural conditions are subject to this type of corrosion destruction. In recent decades, this problem has become of paramount importance. It is difficult to find an industry that does not deal with atmospheric corrosion and protection against it. The global cost of atmospheric corrosion exceeds USD 100 million per year [1]. In this regard, the development of methods for the protection against atmospheric corrosion is a very urgent problem, both from a scientific, technical, and economic point of view.

It is especially important to inhibit the corrosion of the most common structural metal alloy, carbon steel, and it is equally important to reduce the corrosion rate of non-ferrous metals such as copper, aluminum, and zinc, since the runoff of corroded metal ions into the environment is undesirable, since this is harmful to the environment [1].

Organic silanes of the general formula RnSi(0R’)4-n are compounds capable of being adsorbed on a metal surface, bonding strongly with it through Me–O–Si bonds [2], and forming thin polymeric self-assembled organosilicon layers. These layers can change the properties of the surface, in particular, by inhibiting the corrosion of metals [3], especially such metals as iron [4], steel [5,6], aluminum [7,8], zinc [9], copper and alloys based on it [10,11,12], and other metals [13,14,15]. Despite the studies devoted to the inhibition of electrolytic corrosion by surface organosilicon layers [3], there are no data in the literature on their effect on the atmospheric corrosion of metals; therefore, it is necessary to study the effect of surface organosilicon layers on the corrosion of metals in the atmosphere.

Before commissioning a metal structure, a chromate conversion coating is applied as a sublayer to the metal surface, before applying a paint and/or polymeric coating to increase anticorrosion resistance and to improve adhesive interactions with the paint or polymer coating at the metal–polymer interface.

However, hexavalent chromium compounds, which are the basis of such coatings, are toxic and even carcinogenic, which has led to the prohibition of their use in developed countries. Therefore, in recent decades, the scientific and technical problem of replacing these chromate coatings with environmentally friendly, but no less effective intermediate coatings, has become acute.

It is believed [16] that environmentally friendly organosilicon surface layers obtained as a result of the adsorption of environmentally safe organosilanes on metal can replace toxic and prohibited chromate coatings.

Since silicone surface layers are expected to prevent the atmospheric corrosion of metals, and given that they are considered as an alternative to toxic chromates often used in atmospheric conditions, the aim of this work was to study the effect of silicone surface layers on metal atmospheric corrosion.

## 2. Materials and Methods

Full-scale natural corrosion tests were carried out on rectangular metal samples with a size of 100 × 150 mm (4 × 6 inches) in accordance with [17], made of carbon cold-rolled steel (grade 08ps, mass content of C = 0.05–0.11%), zinc (grade Z0, mass content of Zn = 99.975%), aluminum (grade A5M, mass content of Al = 99.95), and copper (grade M1, mass content of Cu = 99.96%).

The preliminary preparation of samples before exposure consisted of mechanical surface cleaning on a grinding machine (marking of sandpaper P1000, grain size of the grinding skin, 14–20 microns), degreasing with ethyl alcohol.

Table 1 shows the compounds used in this study to modify the surface of metals.

Modification of the surfaces of the samples was carried out by completely immersing the metal plates in an aqueous or organic solution of modifiers for 10 min. After modification, the samples were immersed in a solvent for 1 min to remove excess modifier, then dried in air.

Both mono- and bicomponent systems were used in this study: single-component solutions in water of 1% VS, 3% VS, 1% AS, 2% AS, and 1% DAS, and single-component solutions in ethyl alcohol of 1% MS, 1 mM BTA, and 10 mM BTA; and a two-component mixture solution [1% VS + 1% AS] in water and a solution of the mixture [1% VS + 1 mM BTA] in ethyl alcohol.

All solutions were prepared with reagents of the brand “Chemically pure” in distilled water.

Full-scale natural corrosion tests were carried out in accordance with the methodology in [17] in a louvered booth, at the testing site of the Moscow Corrosion Station of the Frumkin Institute of Physical Chemistry and Electrochemistry, RAS, located in the southwest of the city of Moscow.

According to [18], the corrosion category of the atmosphere at the test site was C2 (low) in relation to carbon steel and aluminum, and C3 (medium) in relation to zinc and copper. The samples were fixed on test benches according to the recommendations in [17]. Five samples were put to the test for each type of metal for all the studied systems (unmodified and modified).

At the end of the one-year full-scale tests, a visual inspection of the samples, their photography, and weighing (after removing the corrosion products) were performed for the subsequent assessment of the degree of corrosion by corrosion testing. The corrosion rate of metal, K (mm/year), after full-scale tests, was determined by the Formula (1):K = ΔL/τ(1)
where ΔL is the change in the thickness of the sample (mm) calculated by (2) and τ is the duration of the tests (year).
ΔL = Δm/ρ(2)
where ρ is the density of the metal (g/mm^3^) and

Δm is the mass loss of the sample per unit area (g/mm^2^) (3):Δm = (m_0_ − m_i_)/S(3)
where m_0_ is the mass of the sample before testing (g),

m_1_ is the mass of the sample after testing and removal of corrosion products (g), and

S is the surface area of the sample (mm^2^).

The effectiveness of metal corrosion inhibition was evaluated by the value of the braking (inhibiting) coefficient (4) [19]:γ = K/K_ing_(4)
where K is the corrosion (rate) of the metal without inhibition and K_ing_ is the corrosion of the metal when using corrosion inhibitors.

The higher the magnitude of γ, the more effective the corrosion inhibition.

Immediately after removing the samples from the exposure and after removing the corrosion products from their surface, the surface condition was assessed using optical microscopy. An optical microscope (Carton SPZT50 (magnification ×200)) was used in this study. The enlarged image of the surface area was recorded using an Amoyca AC-300 digital CMOS video camera connected to the eyepiece via an adapter. The resolution of the camera was 2048 × 1536 dpi. The data from the camera were processed in the Corel PHOTO-PAINT X3 graphics editor.

Measurements of the values of the metal corrosion potential (E_cor_) were carried out in a standard three-electrode cell using an IPC-Pro MF potentiostat. The samples were stripped with sandpaper of grade “0”, degreased with alcohol, and then were additionally washed in an ultrasonic bath, “Sapphire—0.8 TC”, in a mixture of C_2_H_5_OH:C_7_H_8_OH = 1:1 for 25 min. In order to exclude the influence of edge effects on the ends of the samples after modification and air drying for 120 min, the samples were coated with a chemically resistant varnish, leaving an “open window” so that the working surface area of the electrode was 1 cm^2^. The measurements were carried out in a borate buffer solution (0.4 M H_3_BO_3_ + xM Na_2_B_4_O_7_) with a pH of 6.7, with the addition of 0.1 M NaCl. All the potential values are given relative to the normal hydrogen electrode (n.h.e.).

## 3. Results and Discussion

The influence of surface self-assembled polymer organosilicon layers on the electrochemical behavior of metals was determined. Table 2 shows the E_cor_ values of the samples of all of the metals studied in this work after the sample was immersed in the solution during 180 s [20].

It is known [20] that under conditions of oxygen depolarization, the displacement of the metal corrosion potential (E_cor_) into the anode region (in positive direction) indicates the transfer of the metal to a passive state and implementation of corrosion inhibition conditions. The obtained results confirmed the above assumption, that the modification of the metal surface by organosilane-based compositions may lead to corrosion inhibition, which manifested in an increase in the potential [20].

The accelerated corrosion tests carried out earlier showed a positive effect on the short-term protection of steel from corrosion by means of modification with organosilanes. In this work, we also carried out long-term corrosion tests.

The results of the one-year field tests showed that preliminary modification of the metal surface with organosilane-based compositions led to the inhibition of metal corrosion in an urban atmosphere. Figure 1, Figure 2, Figure 3 and Figure 4 show the data on the corrosion rates of the metals after the one-year exposure of the metal samples in a louvered booth, obtained by the gravimetric method. It was observed from the figures that the preliminary modification of the metal surface by silane-based compositions led to a decrease in the rate of atmospheric corrosion. The surface layers formed during the surface modification with solutions of two-component mixed compositions (a mixture of VS–AS and VS–BTA) most effectively inhibited corrosion (Table 3).

It is shown that the corrosion of metals in the conditions of an urban atmosphere, as expected [1], has a local character. Visual inspection of the samples showed the presence of two types of corrosion defects on the metal surface (cavities and pittings), traces of which were fixed after the removal of the corrosion products. In order to study this phenomenon in more detail, an optical microscopic examination of the metal samples was performed at the end of the one-year field tests. Namely, a detailed optical microscopic study of copper exposed in an urban atmosphere for one year was carried out.

The micrograph of the copper surface area after one year of full-scale corrosion tests (Figure 5a) shows that the metal surface was uniformly covered with a dense layer of corrosion products. After removing them, there were “stripping lines”, which were traces formed on the surface after polishing it (when preparing the samples for testing) and corrosion defects, also known as pittings. On the unmodified copper surface, small defects were concentrated inside the “stripping lines”, filling them tightly. Larger defects were also observed on the relatively smooth areas between the “stripping lines” (Figure 5b). The average diameter of such defects was about 4.5 microns. An examination of the copper samples, pre-modified with 1% VS solution after exposure during one year in an urban atmosphere, showed that in this case, the metal corroded locally. The entire surface of the sample was covered with small defects, the average diameter of which did not exceed three microns. Thus, the preliminary modification of the copper surface with a 1% solution of VS led to the inhibition of the atmospheric corrosion of copper, but the inhibitory effect was small. The values of the corrosion inhibition coefficients were: 1.79, in the case of uniform corrosion (Table 2), and 1.5, for local (optical microscopy) corrosion.

Increasing in the concentration of the modifying solution of VS to 3% or changing the modifier (from vinylsilane to aminosilane) did not prevent the localized corrosion of copper.

An inspection of the samples modified with 3% VS solution after testing showed the presence of pitting on the surface. Moreover, the smaller pitting were located on the “stripping line”, while the larger ones were on the “smooth” surface (between the lines) of the samples. The value of the average diameter of the pitting was 3.5 microns. An inspection of the samples modified with solutions based on aminosilane (AS) showed that after one year of full-scale tests, the surface was covered with pittings, the average diameter of which was about four microns, and the defects were located both on the “stripping lines” and on the “smooth” surface.

Thus, the modification of the copper surface by single-component solutions of silane-based compositions led to some inhibition of the atmospheric corrosion of copper. The most effective of the used solutions was a modifying solution based on a 1% vinylsilane solution; however, the inhibitory effect was small and inferior to the effect of traditionally used corrosion inhibitors [20].

In addition to modifying the surface with single-component solutions, the preliminary modification of the surface with solutions containing two-component mixtures was also performed. Carrying out an optical microscopic examination of the surface of copper, modified with solutions of bicomponent mixtures after one year of full-scale corrosion tests, showed that there were pittings on the surface; however, their number was significantly less than in the cases of unmodified and modified metal, and in the case of single-component modifying solutions. In addition, after the modification with a solution of a two-component mixture, pitting was registered only on the “stripping lines” and was not observed on the “smooth” surface. The average diameter of the pitting was less than two microns. Apparently, modification of the surface by a mixture of vinyl- and amino-containing silanes led to a stronger binding of the silane layer to the surface, and the formation of a more tightly cross-linked surface layer, since it is believed [21] that amino compounds are a catalyst for both the condensation of silane molecules with surface hydroxyl groups [22] (reaction (5)) and condensation (polymerization) of neighboring adsorbed molecules containing silanol groups (reaction (6)) [22,23]. Moreover, amino-containing silanes, and in particular, aminopropyltriethoxysilane, are self-catalysts when bound to a hydroxylated surface [22,23].
(5)$$Me-OH+OH-\begin{array}{c} |\\ Si\\ | \end{array}R\begin{array}{c} -H_{2}O\\ \to \end{array}Me-O-\begin{array}{c} |\\ Si\\ | \end{array}R$$
(6)$$n\left\{Me-O-Si{\left(R\right)}_{3}\right\}\to Me|-{\left[O-\begin{array}{c} R\\ |\\ Si-\\ | \end{array}\right]}_{n}$$

It should be noted that in a mixture (for example, with an organosilane), the effectiveness of a corrosion inhibitor increases significantly. Compounds of the azole class are effective corrosion inhibitors of non-ferrous metals [21,24,25,26,27,28,29,30,31,32,33]. To study the effect of mixtures of organosilanes and corrosion inhibitors on atmospheric corrosion, a mixture of vinylsilane with benzotriazole (BTA) was used. In addition to the expected increase in the effectiveness of the corrosion inhibitor when mixing it with silane, it was assumed that a more tightly cross-linked surface layer would form when the metal was modified with such a mixture, due to the reaction of condensation of the silanol and amino groups [19] with the formation of Si–N bonds (reactions (7) and (8)) in the volume of the surface layer, additionally forming bonds other than siloxane (Si–O–Si) bonds.
(7)$$OH-\begin{array}{c} |\\ Si\\ | \end{array}-\begin{array}{c} |\\ C\\ | \end{array}-R_{1}+H\begin{array}{c} |\\ N \end{array}-R_{2}\to -\begin{array}{c} |\\ Si\\ |\\ HNR_{2} \end{array}-\begin{array}{c} |\\ C\\ | \end{array}-R_{1}+H_{2}O$$
(8)$$R\begin{array}{c} |\\ Si\\ | \end{array}-OH+H\begin{array}{c} |\\ N \end{array}-R^{\prime }\begin{array}{c} -H_{2}O\\ \to \end{array}R-\begin{array}{c} |\\ Si\\ | \end{array}-\begin{array}{c} |\\ N \end{array}-R^{\prime }$$

Reaction (7) proceeded in parallel with reaction (6), leading to the formation of a larger number of cross-linking bonds in the surface film. As a result, a silicon–azole layer was formed on the surface with a higher degree of cross-linking than in the case of a single-component composition (and where only reaction (6) took place), and prevented the penetration of aggressive components of the medium to the surface, having improved blocking characteristics and the capability of more effectively inhibiting corrosion.

In previous works, we studied in detail the structure of layers formed on the surface of metals upon modification of their surface with organosilanes, with in particular, vinylsilane [10,34,35], BTA [36], and their mixture [6,26]. We found that a surface polymer-like layer was formed on the metal, cross-linked inside by siloxane (Si–O–Si) and siloxane–azole (Si–N) bonds. The scheme of such a layer is presented in Figure 6.

As expected, the mixture of VS and BTA inhibited the local corrosion of copper. So, after one year of exposure, the average diameter of the pittings on the copper modified by a solution of a mixture of VS and BTA decreased by almost three times, compared with the unmodified copper, and almost two times, compared with the modification with a solution of only VS.

Thus, the modification of the copper surface with a solution of a two-component mixture based on organosilanes led to the inhibition of pitting during the atmospheric corrosion of copper. Full-scale natural corrosion tests of zinc modified with organosilane-based solutions were carried out. Figure 3 shows the results of the gravimetric measurements of zinc corrosion rates after one year of full-scale corrosion tests in an urban atmosphere. It was observed (Figure 7) that the preliminary modification of the surface with organosilane-based compositions, for all the studied compositions, contributed to the inhibition of the atmospheric corrosion of zinc. The maximum inhibition efficiency was achieved, as in the case of copper, after modification with solutions of two-component mixtures (Table 3).

Optical microscopic studies of zinc samples after the corrosion tests showed that zinc, under the influence of an urban atmosphere, such as copper, corroded locally. Large defects were found on the surface of the unmodified zinc (from 10 to 30 microns in the amounts of 10 pcs./0.5 mm^2^) (Figure 7a).

After the removal of the corrosion products from the surface, traces of caverns with geometric dimensions from 10 to 30 microns were mainly observed on it (Figure 7b).

Optical microscopic examination did not reveal a significant effect of the preliminary modification by single-component silane solutions on the local corrosion of zinc. Figure 8 shows micrographs of the zinc samples pre-modified with a 1% aqueous solution of VS after one year of full-scale natural corrosion tests in an urban atmosphere. Thus, after modification of the zinc surface, corrosion defects were observed on it after full-scale tests (Figure 8), the number of which was comparable to an unmodified sample, and the size decreased by only 30 microns. After the removal of the corrosion products, as well as in the case of the unmodified metal (Figure 7b), traces of corrosion caverns were found (Figure 8b), their geometric dimensions of which varied from 5 to 20 microns.

After one year of tests of the zinc samples previously modified with a 3% VS solution, traces of corrosion defects (pitting and caverns) were also found on the surface. The geometric dimensions of the defects found on the surface after the removal of the corrosion products ranged from 10 to 15 microns.

Optical microscopic examinations of the zinc samples, pre-modified with solutions of a two-component mixture based on organosilanes, showed that when using a mixture solution (1% VS and 1% AS) after one year of field tests, only rare associations of small pittings were found on the zinc surface (Figure 9a), the sizes of which varied from 3 to 5 microns; i.e., the geometric dimensions of the defects formed in this case were several times smaller than for unmodified metal or for samples modified with a single-component solution. The number of defects was also significantly less.

The study of the zinc samples pre-modified with a solution of the two-component mixture of [VS + BTA] showed that after one year of corrosion tests, the surface was covered with a layer of corrosion products, under which there were corrosion defects (pittings with a diameter of 3–5 microns and caverns with a diameter of 7–15 microns) (Figure 10a,b). Thus, in the case of the preliminary modification of the surface of zinc, with a solution of a two-component mixture of organosilane and a corrosion inhibitor after one year of exposure to an urban atmosphere, corrosion defects were registered on the surface. Their number and size were slightly smaller than in the case of the exposure of unmodified zinc or zinc modified by a single-component solution of organosilane, which indicates the inhibition of the local atmospheric corrosion of zinc.

Optical microscopic examination of the surfaces of the aluminum and carbon steel after the annual field tests showed that the surface self-assembled organosilicon layer formed during modification of the metal with a binary mixture solution, as well as in the cases of copper and zinc, inhibiting local atmospheric corrosion. For example, for aluminum (Figure 11 and Figure 12), on the unmodified metal and after modification with a mixture of 1% VS + 1 mM BTA, after the corrosion tests, pittings of approximately the same average size were found on the surface (the diameter of the average defect was about 6.5 microns). However, the size of the pitting on the aluminum sample after modification with a binary mixture was more than three times less than on the surface of the unmodified one. Namely, the number of pittings per 0.5 mm^2^ was: twenty pcs. and nine pcs. for unmodified aluminum (Figure 11) and aluminum modified with a mixture of VS + BTA (Figure 12), respectively. After modification of the surface with a one-component solution (1% VS), 23 pittings were formed.

The study of unmodified carbon steel showed that the surface was subject to uniform corrosion, against which deep defects were found with an average diameter of 15–20 microns (Figure 13), with a depth of 7–10 microns.

In the case of modification of the steel surface with single-component silane solutions (1% VS and 3% VS), as well as a mixture of [VS + BTA], some inhibition of local corrosion was observed (Figure 14); namely, the average diameter of the pitting decreased to 9–10 microns and the depth, to four microns. The best inhibitory effect was observed after steel surface modification with a mixture of vinyl- and aminosilanes (1% VS + 1% AS) (Figure 15), since the minimum number of local corrosion defects was recorded on the surface.

Thus, the preliminary modification of metals with solutions of two-component mixtures (namely, a mixture of vinylsilane with aminosilane and benzotriazole) led to the inhibition of both uniform and local corrosion caused by the action of an urban atmosphere.

## 4. Conclusions

Full-scale corrosion tests of structural metals were carried out for steel, copper, zinc, and aluminum, modified with organosilane-based compositions, in an urban atmosphere.After one year of natural corrosion tests, it was found that preliminary modification of the metal surface with organosilane-based compositions led to the inhibition of both uniform and local corrosion of the metals.The greatest inhibitory effect was demonstrated by the mixed two-component modifying compositions: mixtures of vinylsilane with aminosilane and vinylsilane with benzotriazole.

## Figures and Tables

**Figure 1 polymers-14-04428-f001:**
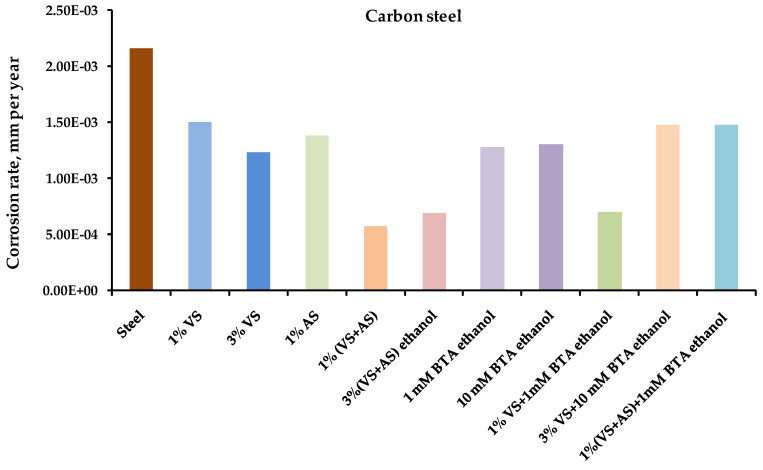
Atmospheric corrosion rates (mm per year) of **carbon steel** modified by organosilane-based compositions. Full-scale, one-year corrosion tests, carried out in an urban atmosphere. Gravimetric assessment of corrosion magnitude.

**Figure 2 polymers-14-04428-f002:**
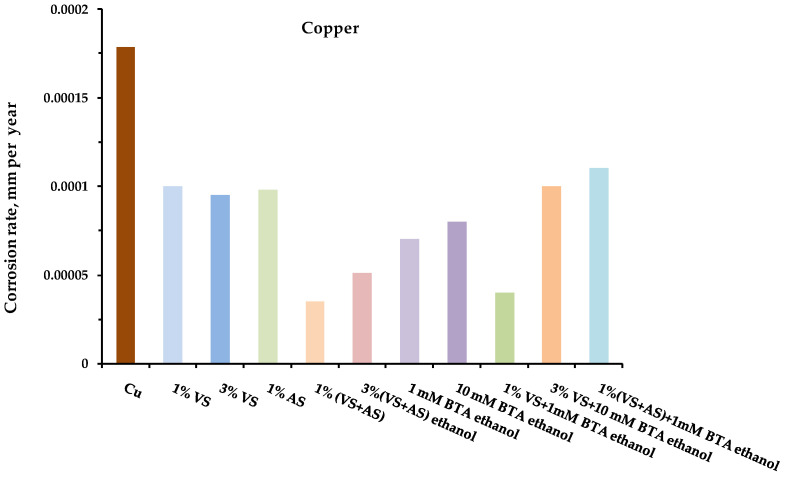
Atmospheric corrosion rates (mm per year) of **copper** modified by organosilane-based compositions. Full-scale, one-year corrosion tests, carried out in an urban atmosphere. Gravimetric assessment of corrosion magnitude.

**Figure 3 polymers-14-04428-f003:**
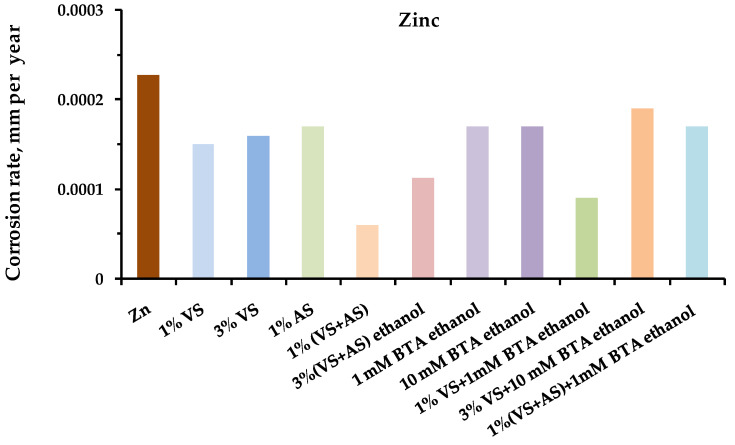
Atmospheric corrosion rates (mm per year) of **zinc** modified by organosilane-based compositions. Full-scale, one-year corrosion tests, carried out in an urban atmosphere. Gravimetric assessment of corrosion magnitude.

**Figure 4 polymers-14-04428-f004:**
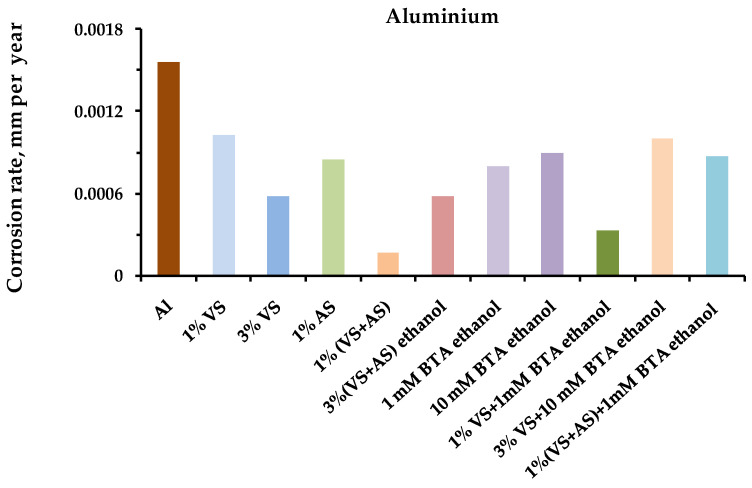
Atmospheric corrosion rates (mm per year) of **aluminum** modified by organosilane-based compositions. Full-scale, one-year corrosion tests, carried out in an urban atmosphere. Gravimetric assessment of corrosion magnitude.

**Figure 5 polymers-14-04428-f005:**
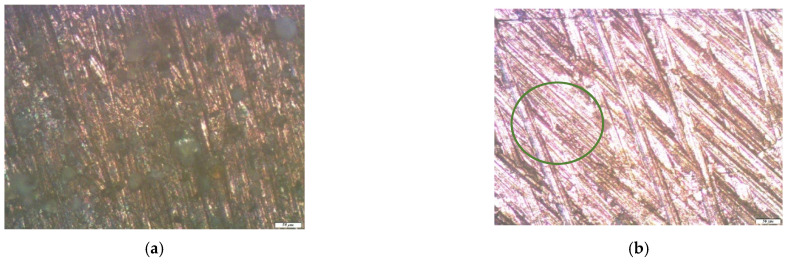
Micrographs of the surface area of an unmodified copper sample after conducting field tests in an urban atmosphere: (**a**) immediately after the tests but before the removal of the corrosion products; (**b**) after the removal of the corrosion products, at magnification 10×, where the green circle marks the area with pitting for a more detailed consideration; and (**c**) after enlarging the surface area (highlighted in (**b**)), at magnification 20×, where the red circles or ovals highlight the corrosion defects (pittings).

**Figure 6 polymers-14-04428-f006:**
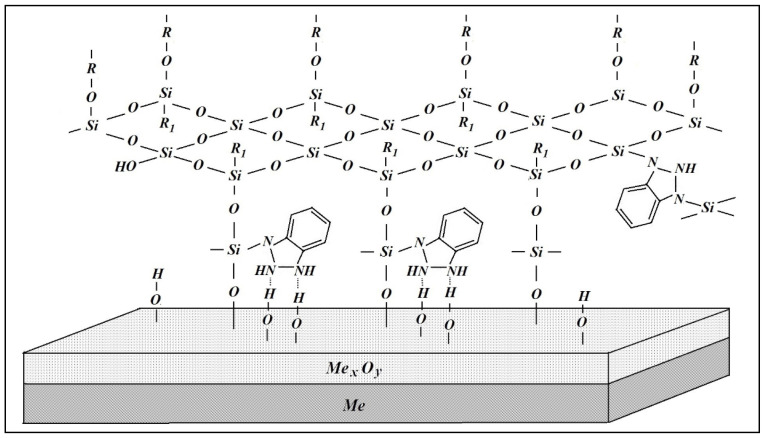
Scheme of the structure of the interface metal/siloxane–azole layer.

**Figure 7 polymers-14-04428-f007:**
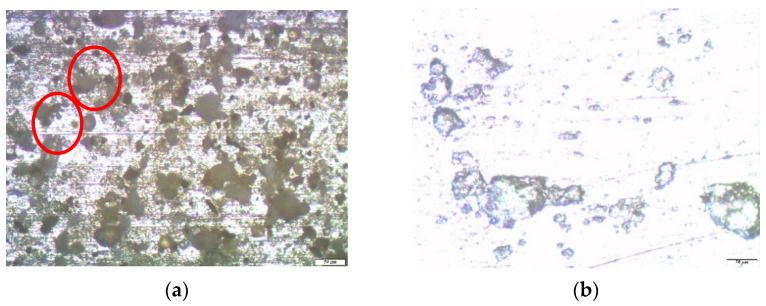
Micrographs of the surface area of an unmodified zinc sample after conducting one-year field tests in an urban atmosphere: (**a**) immediately after testing but before the removal of corrosion products, with hemispherical pitting marked in red; and (**b**) after the removal of corrosion products.

**Figure 8 polymers-14-04428-f008:**
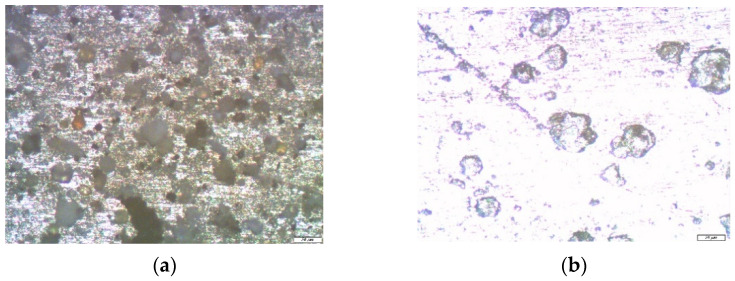
Micrographs of the surface area of the zinc sample, pre-modified with a 1% VS solution, after conducting full-scale, one-year natural tests in an urban atmosphere: (**a**) immediately after testing but before the removal of the corrosion products; and (**b**) after the removal of the corrosion products.

**Figure 9 polymers-14-04428-f009:**
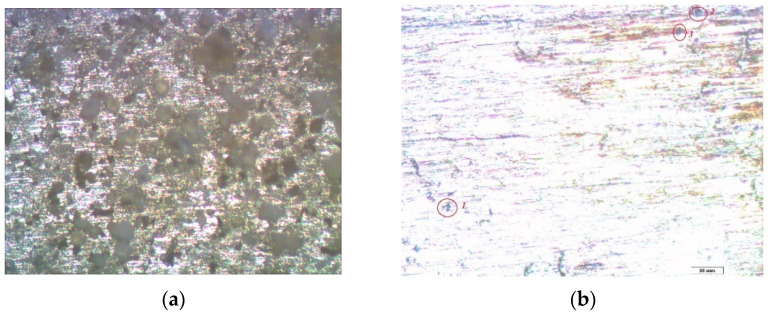
Micrographs of the surface area of a zinc sample pre-modified with a solution containing a mixture of [1% VS + 1% AS] after conducting full-scale, one-year natural tests in an urban atmosphere: (**a**) immediately after the tests but before the removal of the corrosion products; and (**b**) after the removal of the corrosion products. Optical magnification of 10×, with defects highlighted in red.

**Figure 10 polymers-14-04428-f010:**
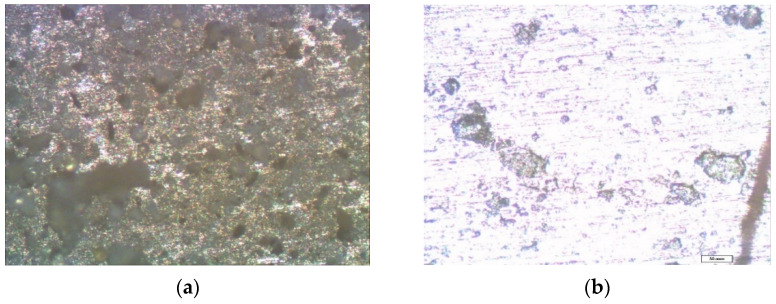
Micrographs of the surface area of a zinc sample pre-modified with a solution containing a mixture of [1% VS + 1 mM BTA], after conducting full-scale, one-year natural tests in an urban atmosphere: (**a**) immediately after testing but before the removal of the corrosion products; and (**b**) after the removal of the corrosion products. Optical magnification of 10×.

**Figure 11 polymers-14-04428-f011:**
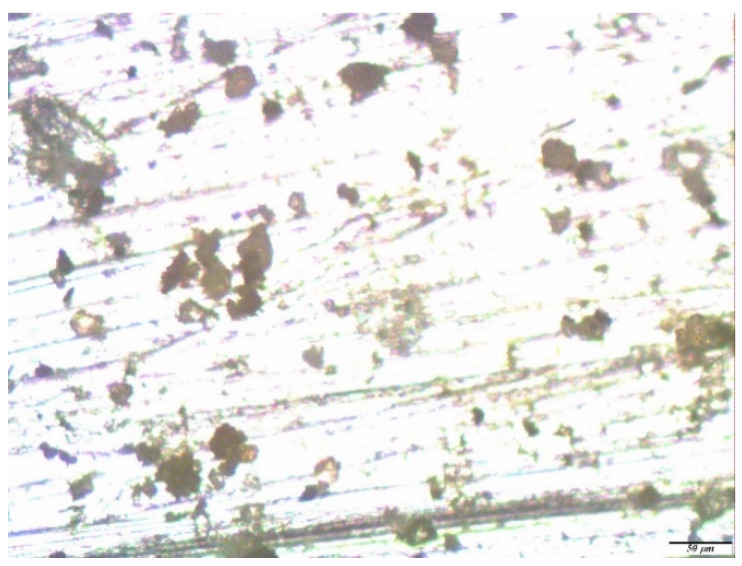
Micrograph of the surface area of an unmodified aluminum sample after conducting full-scale, one-year tests in an urban atmosphere, after the removal of the corrosion products.

**Figure 12 polymers-14-04428-f012:**
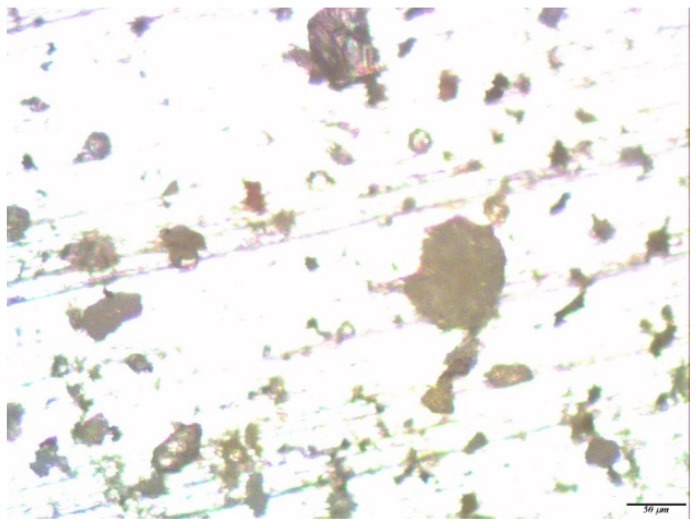
Micrograph of the surface area of an aluminum sample modified with a solution of a mixture of [1% VS + 1 mM BTA], after conducting full-scale, one-year natural tests in an urban atmosphere and removing the corrosion products.

**Figure 13 polymers-14-04428-f013:**
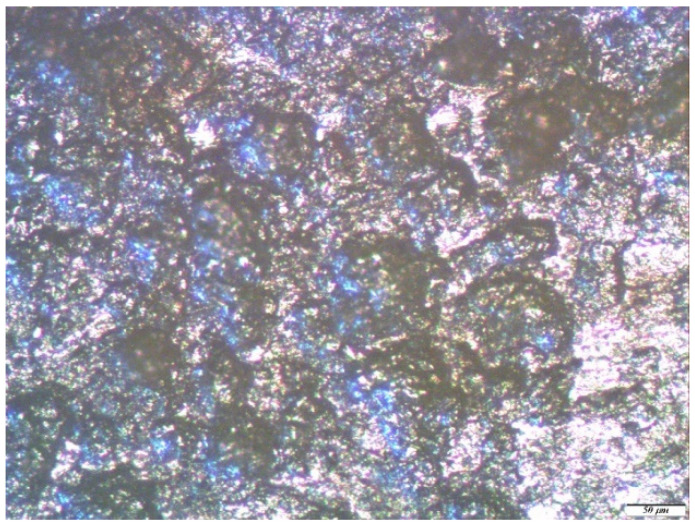
Micrograph of the surface area of an unmodified steel sample after the removal of the corrosion products.

**Figure 14 polymers-14-04428-f014:**
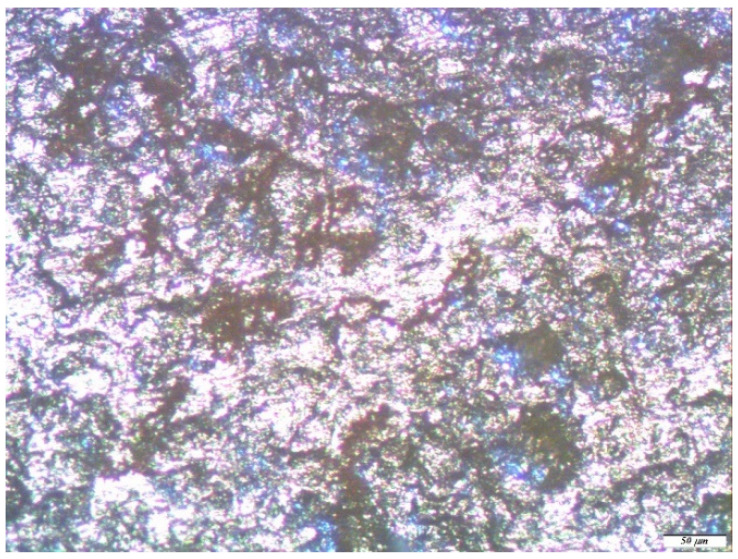
Micrograph of the surface area of a carbon steel sample, modified with a 1% VS water solution or 3% VS water solution, after conducting full-scale, one-year natural corrosion tests in an urban atmosphere and removing the corrosion products.

**Figure 15 polymers-14-04428-f015:**
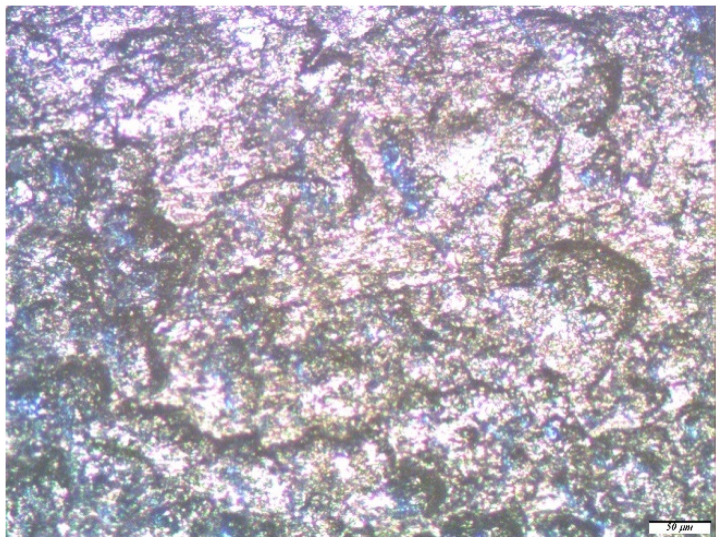
Micrograph of the surface area of a carbon steel sample, modified with solution of a mixture [1% VS + 1% AS], after conducting full-scale, one-year natural corrosion tests in an urban atmosphere and removing the corrosion products.

**Table 1 polymers-14-04428-t001:** Modifier compounds used on the surfaces of metals.

No.	Conditional Designation	Title	Chemical Formula
1	VS	Vinyltrimethoxysilane	CH_2_=CH–Si(OC_2_H_5_)_3_
2	AS	γ-Aminopropyltriethoxysilane	NH2(CH_2_)_3_–Si(OC_2_H5)_3_
3	DAS	Aminoethylaminopropyltrimethoxy silane- Diaminsilane	NH_2_–CH_2_–CH_2_–NH–CH_2_–CH_2_–CH_2_–Si(OCH_3_)_3_
4	MS	γ-Methacryloxypropyltrime- thoxysilane	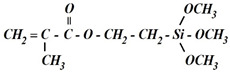
5	BTA	1.2.3-benzotriazole	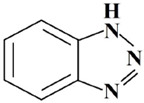

**Table 2 polymers-14-04428-t002:** Values of the corrosion potential (E_cor_) of metal samples without pretreatment of the surface and modified with organosilane-based compositions. Borate buffer (pH 6.7) was used with the addition of 0.1 M NaCl.

No.	System for Modifying the Sample Surface	Ecor, V			
		Copper	Zinc	Aluminum	Carbon Steel
1	Blank	−0.020	−0.835	−0.653	−0.405
2	1% VS	−0.027	−0.834	−0.640	−0.137
3	3% VS	−0.83	−0.770	−0.652	−0.098
4	1% VS + 1% AS	0.048	−0.796	−0.532	−0.078
5	1% VS + 1 mM BTA	0.001	−0.830	−0.453	−0.116

**Table 3 polymers-14-04428-t003:** Inhibition coefficients of uniform corrosion.

Modifying Solution	Inhibition Coefficient of Uniform Corrosion γ (Gravimetric Data)			
	Copper	Zinc	Aluminum	Carbon Steel
Blank	1.00	1.00	1.00	1.00
1% water solution of VS	1.79	1.52	1.52	1.44
3% water solution of VS	1.88	1.43	2.70	1.76
1% water solution of AS	1.82	1.34	1.84	1.57
Alcohol solution of the mixture: [1% VS + 1 mM BTA]	4.46	2.53	4.74	3.09
Water solution of the mixture: [1% VS + 1% AS]	5.10	3.80	9.20	3.79

## Data Availability

Not applicable.

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
