# Peer review of "Effect of Organosilicon Self-Assembled Polymeric Nanolayers Formed during Surface Modification by Compositions Based on Organosilanes on the Atmospheric Corrosion of Metals"

_polymers, 2022, doi:10.3390/polym14204428_

Round 1
Reviewer 1 Report
Manuscript entitled "Effect of organosilicon self- assembled polymeric nanolayers formed during surface modification by compositions based on organosilanes on atmospheric corrosion of metals" . This manuscript brings any new knowledge or data on materials property and therefore only contribution may be in novel preparation method, still this point is not elaborated properly. Presentation and writing is rather poor; there are several statements not supported with data and even some flaws. Some test for the different layer should be given. For these reasons I suggest to reject or major revesion paper in the present form.
1)It is noted that your manuscript needs careful editing by someone with expertise in technical English editing paying particular attention to English grammar, spelling, and sentence structure so that the goals and results of the study are clear to the reader.
2)Also, in the introduvtion section, there are few explanations of the rationale for the study design, please give the present problem of the recent research and your research, thus the necessity of your research shoule be careful explained.
3) The possible mechnism of the vs + bta should be explained,
4)And the surface characterizaion of VA +BTA should be added. For example to confrimed the formation of VA+BTA layer.
Author Response
"Please see the attachment."

Reviewer 2 Report
In this manuscript result studies concerning applications of different functional trialkoxy)silanes (containing vinyl, amino and methacryloxy functionalities) for corrosion protection of metallic surfaces has been desribed. It is continuation or research studies conducted by a team of authors in this field.
Reaction schemes (7) and (8) must be corrected. First, carbon and silicon atoms are four valent, but not two valent. Moreover, reaction scheme (7) seems to be incorrect. Of course a H atom of amine group can form a weak hydrogen bond with O atom of alkoxysilane. I have many doubts, if reaction of alkoxysilane groups with amine group can proceed through such an intermediate state as shown in (7), and a water molecule can not be eliminated at the same time, but later. If silazanes are formed in reaction medium on metallic surfaces, they further hydrolize into Si-OH groups (in the presence of metals as catalysts), which undergo both homocondensation and heterocondenstion reactions with alkoxysilane groups, with formation of resinous silsesquioxane structures.
Author Response
"Please see the attachment."

Reviewer 3 Report
Reviewer Comment for Author:
1- Qualitative information’s are missing in abstract. Abstract should be to improve with more specific results.
2- Minimize keywords and use meaningful words
3- In the introduction, you only talk about corrosion. You should reduce this part and explain what you did in your work
4- Indicate the methods of analysis in the abstract and in the introduction
5- In paragraph 2 materials and methods, it mixes materials and methods; it should be separated and briefly explained
- The molecules must be schematized in the same form (table)
- Name the method you used
6- The stabilization potential of different metals is greater than 60 s (approximately for all 30 min) otherwise cite the references
7- The titles of figures 1, 2, 3 and 4 are larger, it must be reduced
8- We use the inhibition efficiency not Inhibition coefficient, it must be calculated
9- The use of optical microscopy for surface analysis is insufficient to know the elements that exist on the metal surface it is necessary to use SEM / EDX and XPS
10- 10. The summary is very poor
11. The bibliographical study is insufficient, it is necessary to add recent references
12-Gravimetry is a non-electrochemical method, its results are not reliable it is necessary to use other electrochemical method (polarization curve and electrochemical impedance spectroscopy) to enhance your work
Author Response
"Please see the attachment."

Round 2
Reviewer 1 Report
The author has made the corresponding changes to the comments of reviewers.
Author Response
Dear Mr. (Mrs.) Reviewer,
We are grateful for your comments and suggestions. Thank you for your reply!
Kind regards,
the authors:
Maxim Petrunin, Alevtina Rybkina, Tatyana Yurasova and Liudmila Maksaeva
Reviewer 3 Report
checks the corrosion potential after 60 seconds, we never get stabilization after 60 seconds (table)
Author Response
Please see the response to your comments in the attached file.
